# Diagnostic Accuracy of an Offline CNN Framework Utilizing Multi-View Chest X-Rays for Screening 14 Co-Occurring Communicable and Non-Communicable Diseases

**DOI:** 10.3390/diagnostics16010066

**Published:** 2025-12-24

**Authors:** Latika Giri, Pradeep Raj Regmi, Ghanshyam Gurung, Grusha Gurung, Shova Aryal, Sagar Mandal, Samyam Giri, Sahadev Chaulagain, Sandip Acharya, Muhammad Umair

**Affiliations:** 1Department of Radiology, Kathmandu University School of Medical Sciences, Dhulikhel 45200, Nepal; 2Department of Radiology, Institute of Medicine, Tribhuvan University, Maharajgunj 44600, Nepal; 3Department of Radiology, KIST Medical College, Tribhuvan University, Lalitpur 44600, Nepal; 4Department of Electronics and Computer Engineering, Institute of Engineering, Pulchowk Campus, Tribhuvan University, Lalitpur 44600, Nepal; 5Department of Radiology, Columbia University Irving Medical Center, New York, NY 10032, USA

**Keywords:** chest X-ray, deep learning, convolutional neural network (CNN), diagnostic accuracy, artificial intelligence (AI)

## Abstract

**Background:** Chest radiography is the most widely used diagnostic imaging modality globally, yet its interpretation is hindered by a critical shortage of radiologists, especially in low- and middle-income countries (LMICs). The interpretation is both time-consuming and error-prone in high-volume settings. Artificial Intelligence (AI) systems trained on public data may lack generalizability to multi-view, real-world, local images. Deep learning tools have the potential to augment radiologists by providing real-time decision support by overcoming these. **Objective:** We evaluated the diagnostic accuracy of a deep learning-based convolutional neural network (CNN) trained on multi-view, hybrid (public and local datasets) for detecting thoracic abnormalities in chest radiographs of adults presenting to a tertiary hospital, operating in offline mode. **Methodology:** A CNN was pretrained on public datasets (Vin Big, NIH) and fine-tuned on a local dataset from a Nepalese tertiary hospital, comprising frontal (PA/AP) and lateral views from emergency, ICU, and outpatient settings. The dataset was annotated by three radiologists for 14 pathologies. Data augmentation simulated poor-quality images and artifacts. Performance was evaluated on a held-out test set (N = 522) against radiologists’ consensus, measuring AUC, sensitivity, specificity, mean average precision (mAP), and reporting time. Deployment feasibility was tested via PACS integration and standalone offline mode. **Results:** The CNN achieved an overall AUC of 0.86 across 14 abnormalities, with 68% sensitivity, 99% specificity, and 0.93 mAP. Colored bounding boxes improved clarity when multiple pathologies co-occurred (e.g., cardiomegaly with effusion). The system performed effectively on PA, AP, and lateral views, including poor-quality ER/ICU images. Deployment testing confirmed seamless PACS integration and offline functionality. **Conclusions:** The CNN trained on adult CXRs performed reliably in detecting key thoracic findings across varied clinical settings. Its robustness to image quality, integration of multiple views and visualization capabilities suggest it could serve as a useful aid for triage and diagnosis.

## 1. Introduction

Chest radiography (CXR) is one of the most widely performed imaging investigations in clinical medicine due to its speed, affordability, and accessibility [1,2]. It is central to diagnosing a wide range of thoracic diseases, including tuberculosis, pneumonia, pulmonary edema, pneumothorax, lung cancer, interstitial lung disease, and cardiomegaly [3,4,5,6,7]. Over two billion chest X-rays are obtained each year globally, yet timely and accurate interpretation remains a major bottleneck [8]. Accurate interpretation requires trained radiologists, who are scarce in many LMICs; for example, Malaysia has only 3.9 radiologists per 100,000 population, while Tanzania, with 58 million people, has just about 60 radiologists [9,10]. In Nepal, the demand for chest radiographs in emergency, ICU, and outpatient settings exceeds the capacity of available specialists. Even in high-income countries, radiologists face mounting workloads, leading to delays, burnout, and missed findings [11].

Manual CXR interpretation suffers from (i) human error in subtle or overlapping findings, (ii) fatigue-induced inconsistency with high daily caseloads, and (iii) limited specialist availability in remote regions [12,13,14]. Second opinions are known to change diagnoses in up to 21% of cases, underscoring the need for assistive tools that can enhance diagnostic consistency and reduce human error [15].

Deep learning-based CNNs have demonstrated potential for automated interpretation of medical imaging [16]. Unlike costly high-performance systems, efficient CNN architectures can be optimized for low-cost hardware and offline operation, making them suitable for deployment in LMIC contexts [17,18]. Beyond diagnostic accuracy, AI integration may reduce reporting time, energy use, and costs associated with centralized image processing [19].

AI-based CNNs show promise for automating chest radiograph interpretation [18]. While AI models for chest X-ray interpretation have advanced, a significant gap exists in their applicability to real-world LMIC settings. Most existing frameworks are developed on curated, single-view (PA) datasets from high-income settings and lack robustness to the common variabilities in image quality, patient positioning, and equipment found in LMIC hospitals [20]. Moreover, few models are designed for offline, low-power operation, a critical requirement for reliable deployment in areas with unstable internet and limited computational resources [21]. This study directly addresses these limitations by developing a CNN framework specifically optimized for offline use and trained on a hybrid, multi-view dataset that includes the challenging imaging conditions typical of LMIC clinical practice.

Our work makes the following key contributions: (i) We present a lightweight CNN architecture capable of efficient offline inference on standard hospital workstations, eliminating dependence on cloud connectivity. (ii) The model is trained and evaluated on a multi-view dataset (PA, AP, and lateral) from emergency, ICU, and outpatient settings, incorporating extensive augmentation to simulate artifacts (e.g., ECG wires, text labels, rotation, and low exposure), thereby enhancing real-world generalizability. (iii) We implement an interpretable 14-color bounding box system with Weighted Boxes Fusion (WBF) to clearly visualize and differentiate co-occurring pathologies: aortic enlargement, atelectasis, calcification, ILD, infiltration, lung mass/nodule, other lesion (bronchiectasis, hilar lymphadenopathy) mass/nodule, pleural effusion, pleural thickening, consolidation, pneumothorax, lung opacity, fibrosis, and cardiomegaly on frontal and lateral chest radiographs. to improve diagnostic throughput and accuracy in resource-constrained environments. (iv) We demonstrate seamless integration into clinical workflow through PACS compatibility and a standalone web interface, providing a practical pathway for implementation.

This study evaluates the diagnostic performance of a CNN trained on large public datasets (Vin Big and NIH Chest-Xray datasets) and fine-tuned on local CXRs from adults in Nepal, tested on 522 adults (frontal and lateral view X-rays), assessing not only sensitivity and specificity but also its impact on radiologist workflow, and real-world deployment feasibility [22]. The system is deployed through a lightweight, web-based React/NodeJS interface compatible with edge GPUs.

## 2. Literature Review

The application of deep learning for automated chest X-ray interpretation has seen rapid progress. Seminal works utilizing large public datasets, such as CheXNet on NIH ChestX-ray14, demonstrated that CNNs could achieve radiologist-level performance in detecting specific pathologies like pneumonia [23]. Subsequent studies explored more complex architecture and larger datasets like MIMIC-CXR to improve overall diagnostic breadth [24].

A critical focus in the recent literature has been improving model generalizability and clinical utility. Majkowska et al. and Seyyed-Kalantari et al. highlighted the risk of underdiagnosis bias when models trained on data from high-income countries are applied to underserved populations [25,26]. This underscores the necessity for locally relevant training data. Furthermore, the value of incorporating lateral views has been recognized; Hashir et al. quantitatively showed that lateral views can reduce false negatives by providing complementary anatomical information [27].

Recent research has also aimed at practical deployment. For instance, Liong-Rung et al. and Kaewwilai et al. developed AI systems to expedite the diagnosis of critical conditions like dyspnea and tuberculosis, addressing workflow delays [28,29]. A relevant study by Singh et al. demonstrated the performance benefit of multi-view analysis, but was often reliant on a computationally intensive architecture requiring stable infrastructure [30].

Our study builds upon and extends this existing research in several ways. While Singh et al. focused on architectural innovation for multi-view analysis, our primary contribution lies in engineering a complete, end-to-end system optimized for offline, low-resource deployment without sacrificing performance on multi-view data. Unlike models dependent on high-end GPUs or cloud APIs, our lightweight CNN ensures accessibility in settings with fundamental technological constraints. Additionally, by training on a hybrid dataset heavily augmented with LMIC-specific artifacts and implementing a clinician-centric visualization system (color-coded bounding boxes), we bridge the gap between algorithmic performance and practical, interpretable clinical tooling for environments with the greatest need.

## 3. Methods

### 3.1. Study Design and Datasets

This retrospective study developed and evaluated a deep learning system for automated detection and localization of thoracic abnormalities on chest radiographs. The model was pretrained on two public datasets: the Vin Big Chest X-ray dataset (5500 images) and the NIH ChestX-ray dataset (1000 images). Domain-specific fine-tuning was conducted using adult chest radiographs acquired from the Emergency Department, Intensive Care Unit, and Outpatient Clinics of Tribhuvan University Teaching Hospital (TUTH), Nepal, between 1 January 2024 and 1 January 2025.

Inclusion criteria were: (i) age ≥ 18 years, (ii) PA, AP, or lateral digital chest radiographs, and (iii) studies performed in ER, ICU, or OPD. Exclusion criteria were: (i) pediatric patients, (ii) post-operative or post-interventional radiographs with extensive hardware artifacts, and (iii) severely corrupted or non-diagnostic images. Pediatric cases were excluded to avoid anatomical confounding.

Three board-certified radiologists independently annotated all images for 14 predefined thoracic pathologies according to the Fleischner Society Glossary and Radiology Assistant standards. Each image was reviewed by two radiologists, and discordant cases were adjudicated by a third senior radiologist to generate a consensus ground truth. To mitigate overfitting, a combination of data augmentation, L2 regularization, dropout, and early stopping was employed to ensure the model generalized to new data rather than memorizing the training set. To prevent data leakage, strict patient-wise splitting was applied: 70% training, 10% validation, and 20% testing (Figure 1). (Appendix A: CLAIMS Check list).

### 3.2. Preprocessing and Augmentation

All radiographs were resized to 640 × 640 pixels and normalized using a two-step strategy: (i) Min–max normalization to scale pixel intensities to [0, 1]. (ii) Z-score normalization using dataset-level mean and standard deviation.

Extensive augmentation was applied using Albumentations, including random rotations (±15°), horizontal flips, contrast and brightness adjustment, gamma correction, Gaussian noise, Gaussian blur, CLAHE, and sharpening. Medical-specific artifact simulations included overlaying synthetic text labels, ECG wires, and ICU bed-like structures. Bounding box integrity was preserved with minimum visibility ≥0.3 and minimum area ≥1 pixel.

### 3.3. Model Architecture

A computationally efficient, lightweight convolutional neural network (CNN) architecture was developed to enable deployment in resource-constrained clinical environments (Figure 2). The backbone network was a custom Cascade Multi-Resolution Feature Network (CMRF-Net) consisting of ten sequential stages: input normalization using Group Normalization; a Shallow Gradient Extractor with dual 3 × 3 convolutions and LeakyReLU activations; a Dual-Path Expansion Unit comprising parallel convolutional branches with channel fusion; a Hierarchical Aggregation Stack implemented with RepConv blocks; a Progressive Depth Encoder using depth wise separable convolutions with spatial down sampling; a Multi-Scale Bottleneck Cluster incorporating spatial pyramid pooling; a Feature Lift and Redistribution module enabling top–down multi-scale feature fusion; a Secondary Upscale and Multi-Branch Mixing module; a Downscale Reintegration Block; and a Deep Recombination Cluster based on dilated convolutions using atrous spatial pyramid pooling (ASPP). Final predictions were produced through YOLO-style resolution-aligned output heads together with a global image-level classification head, enabling simultaneous multi-scale localization and whole-image abnormality assessment.

The network was trained using an input resolution of 640 × 640 pixels, a batch size of 8 images per GPU, and a total of 300 training epochs. Optimization was performed using the AdamW optimizer (learning rate = 1 × 10^−3^; β_1_ = 0.9, β_2_ = 0.999; weight decay = 5 × 10^−4^). A two-phase learning rate schedule was adopted, consisting of a linear warmup over the first five epochs (initial learning rate = 1 × 10^−7^) followed by cosine annealing decay to a minimum learning rate of 1 × 10^−6^. Gradient clipping (maximum norm = 10.0) and an exponential moving average of model parameters (decay = 0.9999) were applied to stabilize training, and mixed-precision training was enabled using PyTorch 2.7.0 automatic mixed precision. Model optimization employed a multi-scale composite loss computed at three detection resolutions (256 × 256, 128 × 128, and 64 × 64), combining Complete IoU (CIoU) loss for bounding box regression, focal loss for objectness prediction (γ = 2.0, α = 0.5), and binary cross-entropy losses for both multi-label classification and global image-level classification (14 disease classes plus a “no finding” category), with loss weights set to λ_box = 5.0, λ_obj = 1.0, λ_cls = 1.0, and λ_global = 0.5. Post-processing employed Weighted Boxes Fusion (WBF) to refine overlapping detections, and Grad-CAM was used to generate saliency-based visual explanations.

### 3.4. Evaluation Metrics

Image quality was evaluated through expert review and objective metrics. During annotation, radiologists identified images with severe motion blur, exposure issues, or positioning errors. We also used Signal-to-Noise Ratio (SNR) and Contrast-to-Noise Ratio (CNR) to flag potentially inadequate images. Those falling below our quality thresholds were reviewed by a senior radiologist, and diagnostically unacceptable images were excluded. Inter-rater agreement was quantified using Cohen’s kappa (κ). Model performance was then rigorously assessed using comprehensive metrics. For the calculations, a confidence threshold of 0.5 was applied to the model’s predictions. The mean Average Precision (mAP) was calculated at an Intersection over Union (IoU) threshold of 0.5 (mAP@0.5), which is standard for object detection tasks. We conducted a reader study to evaluate clinical utility, measuring the AI system’s integration ease with radiologists’ workflow.
Model Performance ParameterCorresponding FormulaArea Under the ROC Curve (AUC)Calculated as the area under the Receiver Operating Characteristic (ROC) curve.SensitivitySensitivity = TP/(TP + FN)        *TP = True Positive*
        *TN = True Negative*        *FP = False Positive*        *FN = False Negative*SpecificitySpecificity = TN/(TN + FP)*        TP = True Positive*
        *TN = True Negative*        *FP = False Positive*        *FN = False Negative*Mean Average Precision (mAP)mAP=1N∑i=1NAPi, *where* APi *is the average precision for class* i *at an Intersection-over-Union (IoU) threshold of* 0.5.*N = represents the total number of distinct object classes (pathologies)*

### 3.5. Deployment and Clinical Integration

The system was integrated with the hospital PACS and tested in a silent clinical trial on 100 consecutive anonymized studies. Three radiologists used the system for one week and provided structured feedback via 5-point Likert scales.

The model was deployed through an offline-capable React/NodeJS web interface running locally with edge-GPU acceleration. Average end-to-end latency (DICOM retrieval → AI overlay display) was recorded. All AI-generated bounding boxes and confidence scores were exportable as DICOM-compatible overlays, ensuring full interoperability with existing radiology workflows.

## 4. Results

### 4.1. Demographics and Diagnostic Performance

The deep learning model demonstrated good diagnostic accuracy across the 14 thoracic pathologies. Lateral performance metrics were pooled in with frontal view performance metrics. The test cohort comprised 522 adult patients with a median age of 58 years (IQR: 42–68). Males constituted 55% (n = 287) of the population. Regarding view distribution, lateral radiographs accounted for 4% of studies, with the remaining 96% being frontal views (PA or AP). Portable (bedside) examinations represented 15% of all radiographs. The images were acquired from a mix of vendor systems, predominantly Philips, Siemens, and GE.

The system achieved an overall area under the receiver operating characteristic curve (AUC) of 0.86, indicating strong discriminatory ability. Model sensitivity reached 68%, while specificity was maintained at 99%. The mean average precision (mAP) of 0.93 confirmed adequate localization capabilities for the bounding box predictions across all pathology classes (Figure 3 and Figure 4).

The pronounced class imbalance in our dataset (e.g., Calcification: 0.38%, Other Lesion: 0.95%) presented a significant challenge. We employed a class-weighted loss function during training to penalize misclassifications of rare classes more heavily and used targeted augmentation (e.g., copy-paste augmentation for rare findings) on the training set. However, for extremely rare classes like “Other Lesion”, these measures were insufficient to overcome the lack of representative examples, resulting in poor recall (Table 1, Table 2 and Table 3).

### 4.2. Co-Occurrence Detection and Visualization

The model effectively identified singular and multiple co-occurrences of pathologies in one X-ray. The unique 14-color bounding box system provided immediate visual differentiation, enabling rapid interpretation of different pathologies in a single X-ray. Confidence score overlays further enhance utility by allowing prioritization of high-certainty detections during time-constrained readings.

### 4.3. Radiologist Performance Enhancement

The uniquely color-coded detection system for 14 pathologies minimized radiologist confusion by eliminating same-color bounding box overlap, while confidence scores enabled prioritization of high-certainty findings. Integration of both lateral and frontal views further streamlined chest X-ray interpretation workflow (Figure 5).

### 4.4. Robustness and Generalization

The model demonstrated potential for consistent performance across challenging imaging conditions, including poor-quality ICU anteroposterior films and lateral views with artifacts. This suggests acceptable resilience of the system to technical variations that could support deployment in diverse environments where ideal imaging conditions are not always achievable.

### 4.5. Explainable AI (XAI) Analysis

We applied Grad-CAM to highlight image regions influencing model predictions. As shown in Figure 6, the heatmaps consistently align with key anatomical areas, for example, cardiac silhouette in cardiomegaly, lung bases in pleural effusion, and parenchymal lesions in nodules, supporting clinical validity. Combined with our color-coded bounding boxes, this provides a simple two-level interpretability system for rapid localization and transparent decision reasoning.

### 4.6. Deployment and Integration

The system was successfully integrated into the hospital’s PACS, operating seamlessly within existing clinical workflows. Offline functionality was confirmed on standard laptops without internet connectivity, ensuring reliable operation in resource-constrained settings. The exportable DICOM overlay capability maintains compatibility with existing medical imaging infrastructure, facilitating smooth adoption into routine diagnostic processes.

### 4.7. Error Analysis

We conducted an analysis of the model’s errors on the test set to identify common failure modes. The most frequent false positives occurred for ‘Lung Opacity’ and ‘Infiltration,’ often in cases with prominent vascular markings or suboptimal exposure that the model misinterpreted as pathology. The most frequent false negatives were for ‘Pneumothorax’, predominantly involving small, apical, or lateral pneumothoraxes on portable AP films, and for the ‘Other Lesion’ class, where the limited number of examples hindered learning.

## 5. Discussion

AI tools have rapidly advanced chest radiograph interpretation, with CNNs demonstrating high diagnostic accuracy across large datasets such as NIH ChestX-ray14 (Wang et al., 2017) and MIMIC-CXR (Johnson et al., 2019) [24,31]. Studies by Rajpurkar et al. (2017) and Hofmeister et al. (2024) found that deep learning models achieved near-radiologist performance on curated datasets [23,32]. However, their generalizability to low- and middle-income countries (LMICs) remains limited. Most of these frameworks were developed using standardized posteroanterior (PA) films obtained in tertiary hospitals with consistent imaging quality and equipment calibration [20]. In contrast, LMIC hospitals frequently face variability in patient positioning, exposure, and machine type, which substantially affects image quality and limits the applicability of these models in real-world clinical practice [21].

Unlike prior CNN models limited to single-view datasets, our system incorporated multiview (anteroposterior, lateral, and portable) radiographs from diverse clinical environments, replicating real-world imaging diversity and challenges not included in traditional public datasets. Certain pathologies, such as retrocardiac consolidation, mediastinal widening, or small pleural effusions, may only be apparent on the lateral projection, clarifying equivocal frontal findings [33]. Hashir et al. (2020) similarly found that training on lateral images reduced false negatives [27].

Our results demonstrated high average specificity (99%) and precision (90%), indicating reliable identification of normal studies and low false-positive rates. Sensitivity was variable across pathologies (average 68%), with the model performing best on common and well-defined findings such as consolidation, atelectasis, and pulmonary fibrosis, all of which showed F1-scores above 0.90. These pathologies likely benefited from clearer radiographic features and higher representation in the training set. Conversely, rarer conditions such as ILD, aortic enlargement, and other lesions exhibited low sensitivity, reflecting limited training examples and the inherent difficulty of recognizing subtle or heterogeneous imaging patterns, consistent with prior reports (Majkowska 2020; Seyyed-Kalantari et al., 2021) [25,26]. Our model showed high sensitivity for consolidation (96.2%) and atelectasis (97.3%), likely due to their distinct radiographic features and higher prevalence in the training data. Conversely, performance was lower for pneumothorax (sensitivity 75.0%) and “Other Lesion” (sensitivity 0.0%). For pneumothorax, the lower sensitivity may be attributed to: (i) The subtle nature of a small apical pneumothorax, especially on portable AP views; (ii) Potential confusion with skin folds or bullae, leading the model to be conservative; (iii) Relative under-representation compared to more prevalent conditions like consolidation. The “Other Lesion” class, encompassing rare entities like bronchiectasis, had very few positive examples, making it impossible for the model to learn robust features, highlighting a challenge of extreme class imbalance. Co-occurrence analysis revealed clinically plausible associations, such as consolidation with atelectasis or pleural effusion, supporting the model’s capacity to recognize physiologic relationships rather than isolated features.

Studies by Liong-Rung et al. (2022) and Kaewwilai et al. (2025) reported that clinicians often have to await formal radiology reports, which can delay the detection of urgent findings such as pneumothorax, pleural effusion and pulmonary tuberculosis [28,29]. Several images may not be reviewed by a radiologist at all in LMICs, in high-volume settings. Integrating AI systems for rapid, automated screening ensures that these studies are at least evaluated, with high-confidence cases flagged for radiologist review and immediate clinical attention. The use of color-coded bounding boxes and confidence-weighted outputs in our model facilitates transparent, interpretable triage, helping prioritize critical cases and support timely patient care. Similar confidence-driven approaches have proven effective in other screening contexts, such as detection of intracranial hemorrhage [34].

Although Transformer-based models (Chen et al., 2021; Touvron et al., 2020) offer high representational power, they require advanced hardware and stable internet access, conditions often lacking in LMIC settings [35,36,37]. In contrast, our lightweight CNN functions efficiently offline on standard hospital workstations, reducing infrastructure and energy demands and improving scalability.

Limitations: This study was conducted retrospectively at a single tertiary hospital, which may restrict generalizability across regions and imaging devices. Class imbalance, particularly for rare conditions such as ILD and calcification, likely contributed to reduced sensitivity. Pediatric and neonatal populations were not included, despite their high disease burden in LMICs. Additionally, prospective workflow evaluation is needed to determine optimal confidence thresholds and their impact on real-world triage efficiency.

Future Directions: Future studies should focus on multicenter validation across varied LMIC healthcare settings, inclusion of pediatric cohorts, and temporal performance assessment on prospective data. Expanding datasets for low-prevalence conditions and exploring class reweighting or synthetic augmentation may improve detection balance. Incorporating advanced explainability approaches, such as attention-based or information-theoretic visualization, could further enhance clinician trust and integration into routine diagnostic pathways. Furthermore, the integration of Large Language Models (LLMs) could revolutionize AI-assisted radiology. A multimodal LLM could synthesize CNN’s visual findings with patient history from electronic health records to generate preliminary, narrative-style reports, further reducing radiologist burden.

## 6. Conclusions

This study presents a CNN-based system for detecting 14 thoracic pathologies, designed to address the technical challenges of diverse clinical environments seen in everyday practice. Trained on a hybrid, multi-view dataset (including AP, lateral, and portable films) and enhanced through data augmentation, the system demonstrated robust performance on poor-quality radiographs, achieving a mean average precision (mAP@0.5) of 93%. Its interpretability features of color-coded bounding boxes and visible confidence scores enable efficient multi-pathology detection and case prioritization. Deployable via both PACS-integrated and standalone interfaces, the system demonstrates potential as a scalable tool for triage and workflow efficiency, with a foundation for future expansion to rarer conditions.

## Figures and Tables

**Figure 1 diagnostics-16-00066-f001:**
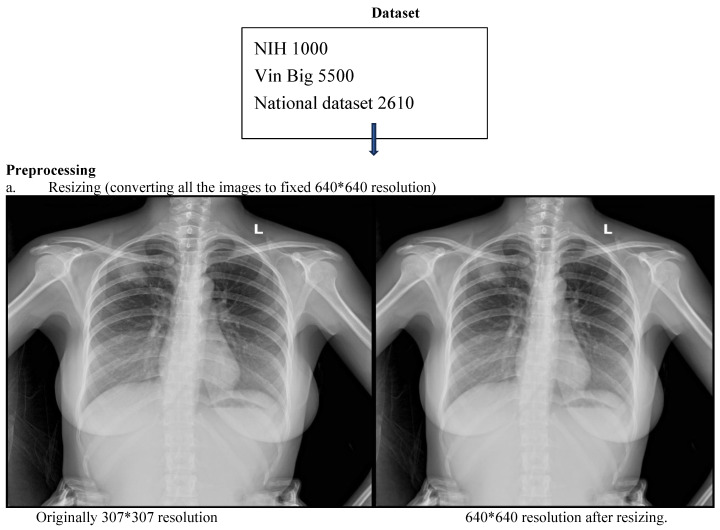
CNN workflow pipeline depicting data sources, preprocessing and bounding box outputs.

**Figure 2 diagnostics-16-00066-f002:**
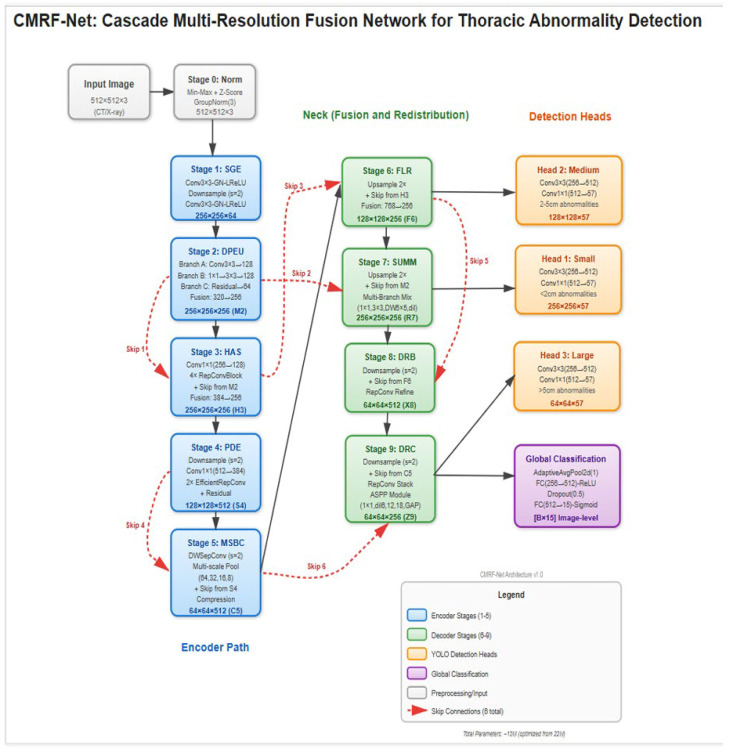
Cascade Multi-Resolution Feature Network (CMRF-Net) architecture. CMRF-Net consists of 10 sequential processing stages with 6 strategic skip connections, designed to capture both fine-grained edge features and high-level semantic patterns across multiple spatial scales. The network follows an encoder–decoder–style cascade, producing three parallel detection heads at 256 × 256, 128 × 128, and 64 × 64 resolutions, along with a global classification head for image-level triage.

**Figure 3 diagnostics-16-00066-f003:**
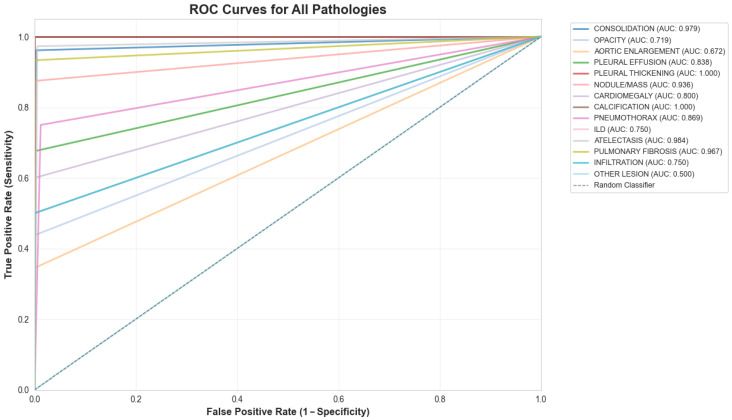
ROC curve for the 14-class model.

**Figure 4 diagnostics-16-00066-f004:**
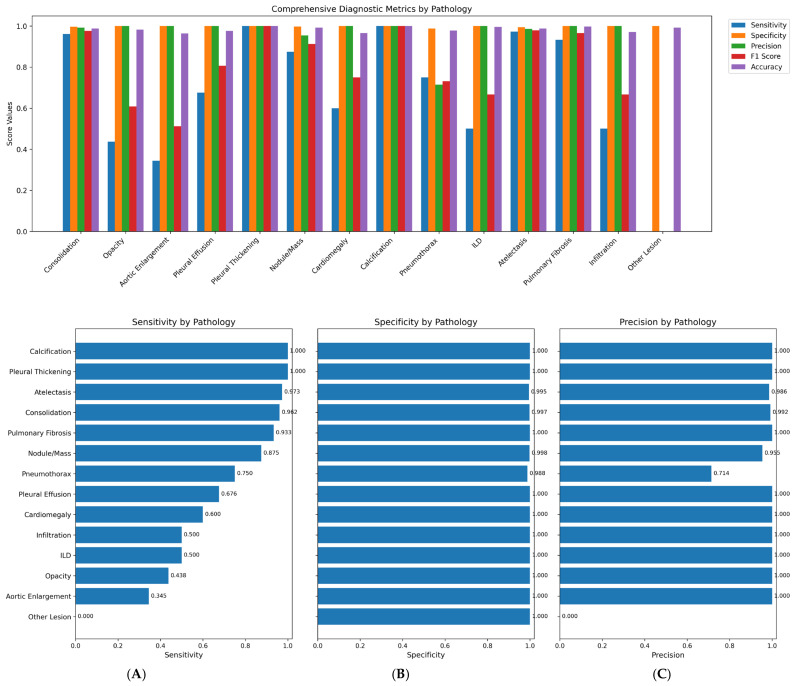
1. Comprehensive diagnostic metrics for 14 classes of pathologies. 2. (**A**): Sensitivity per pathological class, (**B**): Specificity per pathological class, (**C**): Precision per pathological class.

**Figure 5 diagnostics-16-00066-f005:**
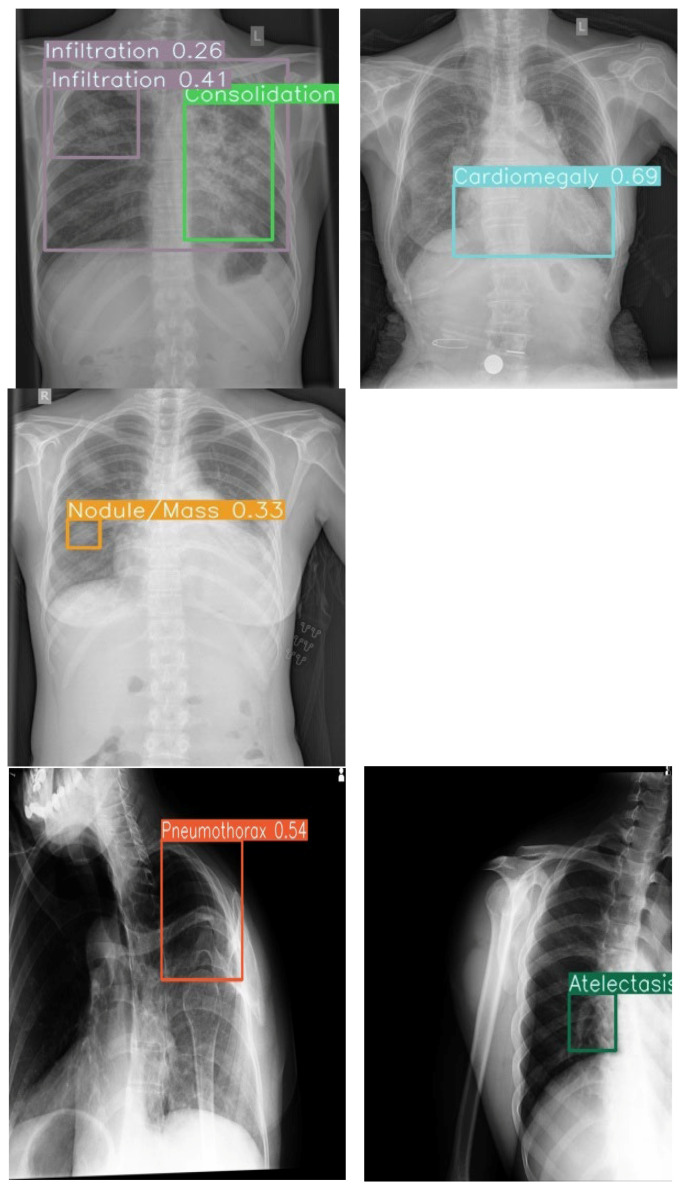
Example outputs with 14-color bounding boxes on multi-pathology cases in Row 1 (Frontal view) and Row 2 (Lateral view).

**Figure 6 diagnostics-16-00066-f006:**
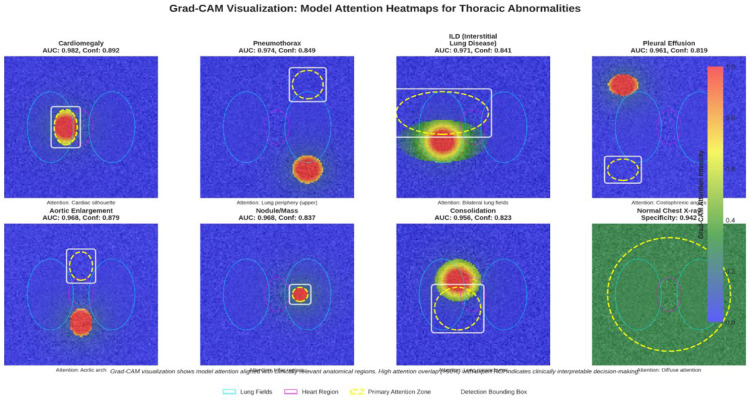
Grad-CAM heatmap overlaid on the X-ray, showing areas of high model activation.

**Table 1 diagnostics-16-00066-t001:** Per pathology prevalence in national test dataset.

Pathology	Positive Cases	Prevalence	Prevalence (%)
Consolidation	155	0.2931	29.31
Opacity	14	0.0265	2.65
Aortic Enlargement	22	0.0416	4.16
Pleural Effusion	33	0.0624	6.24
Pleural Thickening	15	0.0284	2.84
Nodule/Mass	19	0.0359	3.59
Cardiomegaly	36	0.0681	6.81
Calcification	2	0.0038	0.38
Pneumothorax	20	0.0378	3.78
ILD	8	0.0151	1.51
Atelectasis	148	0.2798	27.98
Pulmonary Fibrosis	15	0.0284	2.84
Infiltration	24	0.0454	4.54
Other Lesion	5	0.0095	0.95

**Table 2 diagnostics-16-00066-t002:** Per pathology diagnostic accuracy metrics.

Pathology	Sensitivity	Specificity	Precision	F1_Score	Accuracy	NPV
Consolidation	0.9615	0.9974	0.9921	0.9766	0.9885	0.9874
Opacity	0.4375	1.0000	1.0000	0.6087	0.9828	0.9825
Aortic Enlargement	0.3448	1.0000	1.0000	0.5128	0.9636	0.9629
Pleural Effusion	0.6757	1.0000	1.0000	0.8065	0.9770	0.9759
Pleural Thickening	1.0000	1.0000	1.0000	1.0000	1.0000	1.0000
Nodule/Mass	0.8750	0.9980	0.9545	0.9130	0.9923	0.9940
Cardiomegaly	0.6000	1.0000	1.0000	0.7500	0.9655	0.9636
Calcification	1.0000	1.0000	1.0000	1.0000	1.0000	1.0000
Pneumothorax	0.7500	0.9880	0.7143	0.7317	0.9789	0.9900
ILD	0.5000	1.0000	1.0000	0.6667	0.9962	0.9962
Atelectasis	0.9730	0.9947	0.9863	0.9796	0.9885	0.9894
Pulmonary Fibrosis	0.9333	1.0000	1.0000	0.9655	0.9981	0.9980
Infiltration	0.5000	1.0000	1.0000	0.6667	0.9713	0.9704
Other Lesion	0.0000	1.0000	0.0000	0.0000	0.9923	0.9923
Mean	0.6822	0.9984	0.9034	0.7556	0.9854	0.9859

Mean mAP: 0.93.

**Table 3 diagnostics-16-00066-t003:** Pathological classes co-occurrence detection and their frequency.

Pathology Pair	Co-Occurrence Count	Co-Detection Prevalence	Co-Detection (%)
Consolidation + Atelectasis	21	0.0397	3.97
Consolidation + Pleural Effusion	8	0.0151	1.51
Consolidation + Pleural Thickening	7	0.0132	1.32
Consolidation + Nodule/Mass	6	0.0113	1.13
Consolidation + Cardiomegaly	6	0.0113	1.13
Atelectasis + Pleural Effusion	6	0.0113	1.13
Consolidation + Pulmonary Fibrosis	5	0.0095	0.95
Atelectasis + Pleural Thickening	5	0.0095	0.95
Consolidation + Opacity	4	0.0076	0.76
Atelectasis + Nodule/Mass	4	0.0076	0.76

## Data Availability

The data supporting the findings of this study are available from the corresponding authors due to (privacy, legal and ethical reasons.). The code used in this study is available from the corresponding author due to (privacy and ethical reasons).

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
