# Peer review of "Diagnostic Accuracy of an Offline CNN Framework Utilizing Multi-View Chest X-Rays for Screening 14 Co-Occurring Communicable and Non-Communicable Diseases"

_diagnostics, 2025, doi:10.3390/diagnostics16010066_

Round 1

Reviewer 1 Report

Comments and Suggestions for Authors

The authors proposed a diagnosis using multi view chest xray. The authors has to address the following comments

  1. The abstract should be rewritten in a structured format, clearly outlining the background, objective, methodology, results, and conclusion of the study.

  2. The introduction needs to be revised to highlight the authors’ specific contributions to the field. Clearly state the research gap, motivation, and how this work advances existing knowledge.

  3. The manuscript currently lacks a dedicated literature review section. A comprehensive review of related and recent studies should be included. In particular, the following recent publication can be referenced and discussed for context and comparison:
    https://ieeexplore.ieee.org/stamp/stamp.jsp?arnumber=10938087

  4. The results section could be strengthened by incorporating Explainable AI (XAI) analyses. Including visual and interpretative explanations (such as heatmaps or saliency maps) would enhance the transparency and interpretability of the proposed diagnostic model

Author Response

To,

Reviewer 1

Round 1

Thank you for providing your feedback. We agree with all points raised and have attempted to include them with amendments. The point-by-point response to comments are as follows:

  1. The abstract should be rewritten in a structured format, clearly outlining the background, objective, methodology, results, and conclusion of the study.
  • Thank you for suggesting appropriate structure for abstract. We agree and have accordingly formatted the abstract for clarity.
    A fully structured abstract has been placed in the revised manuscript itself, and also been included in blue highlighted text at bottom of this document.

  1. The introduction needs to be revised to highlight the authors’ specific contributionsto the field. Clearly state the research gap, motivation, and how this work advances existing knowledge.
  • Thank you for the feedback. We appreciate it and have added the above highlights in Introduction section in paragraph 4 and 5 and been included in blue highlighted text at bottom of this document.

  1. The manuscript currently lacks a dedicated literature review section. A comprehensive review of related and recent studies should be included. In particular, the following recent publication can be referenced and discussed for context and comparison:
    https://ieeexplore.ieee.org/stamp/stamp.jsp?arnumber=10938087
  • Thank you for the feedback. We have added a dedicated literature review section after Introduction section and also been included in blue highlighted text at bottom of this document.

  1. The results section could be strengthened by incorporating Explainable AI (XAI) analyses. Including visual and interpretative explanations (such as heatmaps or saliency maps) would enhance the transparency and interpretability of the proposed diagnostic model.
  • Thank you for your feedback. The “Methods” sections, precisely under “Model Architecture” has been equipped with explainable AI (XAI). Under “Results”, explainable AI (XAI) subheading has been added, and image has been added depicting it for greater clarity and also been included in blue highlighted text at bottom of this document.

With thanks and regards,

Authors

Reviewer 2 Report

Comments and Suggestions for Authors

First of all, I would like to thank the authors for their efforts in this manuscript. This manuscript evaluates the Diagnostic Accuracy of an Offline CNN Framework Utilizing Multi-View Chest X-rays for Screening 14 Co-occurring Communicable and Non-Communicable Diseases. Although the overall quality of this manuscript is good, the following comments are intended to improve the overall quality of it:
1. In the Methods section, it is necessary to mention the systems used and their details.
2. Sections 2.3 and 2.5 are discussed in a very general way. It is recommended that these two sections be reviewed in more detail.
3. In the case of the datasets used, more details need to be written. For example, in the case of the local dataset used, you need to state your inclusion criteria. You also need to specify the exclusion criteria.
4. In the Methods section, specify how the image quality was assessed. Was a quantitative measure used?
5. Describe how the radiologists evaluated the data. Were there any specific guidelines for evaluation and naming?
6. It is recommended to show the architecture of your model in a schematic form.
7. In the abstract and main text, please write the full form of abbreviations when first used.
8. How did you deal with biases due to overfitting and data leakage?
9. In the results section, before reporting the accuracy results, please indicate the primary statistical results from the datasets, such as demographics.
10. It is recommended to evaluate, revise, and complete the manuscript according to the CLAIM reporting checklist. Finally, submit the completed checklist as an appendix.
11. In the discussion section, it is recommended to discuss the differences in performance between different diseases. For example, what is the reason for the poor performance in pneumothorax?
12. It is recommended that a discussion be held about the possible role of LLM and XAI in the future of this field.
13. In the method section, it is recommended that the formulas used be mentioned.
14. If generative artificial intelligence was used in the writing of this manuscript, please disclose it.

Author Response

To,

Reviewer 2

Round 1

Thank you for providing your feedback. We agree with all points raised and have attempted to include them with amendments. The point-by-point response to comments are as follows:

  1. In the Methods section, it is necessary to mention the systems used and their details.
  • Thank you for the feedback. Based on other feedback presented below, and other reviewers Methods section has been thoroughly revised to provide more clarity.

“This retrospective study developed an AI model using a hybrid dataset approach. Initial pretraining was performed on two public datasets: the Vin Bigdata Chest X-ray collection (5500 images) NIH Chest X-ray dataset (1000 images). (30) , (31) For domain-specific fine-tuning, we curated a local dataset of adult chest X-rays from Tribhuvan University Teaching Hospital, sourced from Emergency Room, Intensive Care Unit, and Outpatient Department settings between [January 1, 2024 – January 1, 2025]. The inclusion criteria: (i) Patients aged 18 years or older; (ii) Digital chest radiographs in PA, AP, or lateral view; (iii) Studies performed in ER, ICU, or OPD settings and exclusion criteria: (i) Pediatric patients (<18 years); (ii) Post-operative or post-interventional images with surgical hardware causing significant artifacts; (iii) Images with severe corruption preventing reliable annotation was utilized for selection of images in dataset. Pediatric cases were excluded altogether to avoid anatomical confounding.”

  1. Sections 2.3 and 2.5 are discussed in a very general way. It is recommended that these two sections be reviewed in more detail.
  • Thank you for feedback. We have expanded the Methodssection with specific technical details, including hardware, software, and a more detailed description of model architecture and deployment. Please refer to the manuscript for more details and diagram.

Model Architecture

A computationally efficient, lightweight convolutional neural network (CNN) architecture was developed to enable deployment in resource-constrained clinical environments. The backbone network was a custom Cascade Multi-Resolution Feature Network (CMRF-Net) consisting of ten sequential stages: input normalization using Group Normalization; a Shallow Gradient Extractor with dual 3 × 3 convolutions and LeakyReLU activations; a Dual-Path Expansion Unit comprising parallel convolutional branches with channel fusion; a Hierarchical Aggregation Stack implemented with RepConv blocks; a Progressive Depth Encoder using depthwise separable convolutions with spatial downsampling; a Multi-Scale Bottleneck Cluster incorporating spatial pyramid pooling; a Feature Lift and Redistribution module enabling top–down multi-scale feature fusion; a Secondary Upscale and Multi-Branch Mixing module; a Downscale Reintegration Block; and a Deep Recombination Cluster based on dilated convolutions using atrous spatial pyramid pooling (ASPP). Final predictions were produced through YOLO-style resolution-aligned output heads together with a global image-level classification head, enabling simultaneous multi-scale localization and whole-image abnormality assessment.

  1. In the case of the datasets used, more details need to be written. For example, in the case of the local dataset used, you need to state your inclusion criteria. You also need to specify the exclusion criteria.
  • Thank you for your feedback. The inclusion and exclusion criteria are now mentioned in first paragraph of “Methods” section.

“This retrospective study developed an AI model using a hybrid dataset approach. Initial pretraining was performed on two public datasets: the Vin Bigdata Chest X-ray collection (5500 images) NIH Chest X-ray dataset (1000 images). (30) , (31) For domain-specific fine-tuning, we curated a local dataset of adult chest X-rays from Tribhuvan University Teaching Hospital, sourced from Emergency Room, Intensive Care Unit, and Outpatient Department settings between [January 1, 2024 – January 1, 2025]. The inclusion criteria: (i) Patients aged 18 years or older; (ii) Digital chest radiographs in PA, AP, or lateral view; (iii) Studies performed in ER, ICU, or OPD settings and exclusion criteria: (i) Pediatric patients (<18 years); (ii) Post-operative or post-interventional images with surgical hardware causing significant artifacts; (iii) Images with severe corruption preventing reliable annotation was utilized for selection of images in dataset. Pediatric cases were excluded altogether to avoid anatomical confounding.”

  1. In the Methods section, specify how the image quality was assessed. Was a quantitative measure used?
  • Thank you we agree with your feedback. In Methods section, under Evaluation metrics the following text is added for clarity.

“Image quality was evaluated through expert review and objective metrics. During annotation, radiologists identified images with severe motion blur, exposure issues, or positioning errors. We also used Signal-to-Noise Ratio (SNR) and Contrast-to-Noise Ratio (CNR) to flag potentially inadequate images. Those falling below our quality thresholds were reviewed by a senior radiologist, and diagnostically unacceptable images were excluded.”

  1. Describe how the radiologists evaluated the data. Were there any specific guidelines for evaluation and naming?
  • Thank you we agree with your feedback. In Methods section, under Study Design and Datasets the following text has been added.
    “Three radiologists annotated all images for 14 pathologies using standardized definitions (Fleischner Society Glossary of Terms for Thoracic Imaging and the Radiology Assistant: Chest X-Ray). Each image was reviewed by two radiologists, with a third resolving disagreements to create the final consensus ground truth for model development.”

  1. It is recommended to show the architecture of your model in a schematic form.
  • Thank you, it has been done so in Methods section.

Figure 2: Cascade Multi-Resolution Feature Network (CMRF-Net) architecture. CMRF-Net consists of 10 sequential processing stages with 6 strategic skip connections, designed to capture both fine-grained edge features and high-level semantic patterns across multiple spatial scales. The network follows an encoder–decoder–style cascade, producing three parallel detection heads at 256×256, 128×128, and 64×64 resolutions, along with a global classification head for image-level triage.

  1. In the abstract and main text, please write the full form of abbreviations when first used.
  • Thank you for your feedback. It has been corrected.

  1. How did you deal with biases due to overfitting and data leakage?
  • We have added this in Methods section under Study Design and datasets.

“To mitigate overfitting, a combination of data augmentation, L2 regularization, dropout, and early stopping was employed to ensure the model generalized to new data rather than memorizing the training set. To prevent data leakage, strict patient-wise splitting was applied: 70% training, 10% validation, and 20% testing.”

  1. In the results section, before reporting the accuracy results, please indicate the primary statistical results from the datasets, such as demographics.
  • Thank you for your feedback. This has been added in results section. “The test cohort comprised 522 adult patients with a median age of 58 years (IQR: 42-68). Males constituted 55% (n=287) of the population. Regarding view distribution, lateral radiographs accounted for 4% of studies, with the remaining 96% being frontal views (PA or AP). Portable (bedside) examinations represented 15% of all radiographs. The images were acquired from a mix of vendor systems, predominantly Philips, Siemens, and GE.”

  1. It is recommended to evaluate, revise, and complete the manuscript according to the CLAIM reporting checklist. Finally, submit the completed checklist as an appendix.
  • A completed CLAIM (Checklist for Artificial Intelligence in Medical Imaging) checklist is provided as a supplementary appendix to this manuscript.

  1. In the discussion section, it is recommended to discuss the differences in performance between different diseases. For example, what is the reason for the poor performance in pneumothorax?
  • Thank you for your feedback. This text has been added to discussion paragraph 3.

Our model showed high sensitivity for consolidation (96.2%) and atelectasis (97.3%), likely due to their distinct radiographic features and higher prevalence in the training data. Conversely, performance was lower for pneumothorax (sensitivity 75.0%) and "Other Lesion" (sensitivity 0.0%). For pneumothorax, the lower sensitivity may be attributed to: (i) The subtle nature of a small apical pneumothorax, especially on portable AP views; (ii) Potential confusion with skin folds or bullae, leading the model to be conservative; (iii) Relative under-representation compared to more prevalent conditions like consolidation. The "Other Lesion" class, encompassing rare entities like bronchiectasis, had very few positive examples, making it impossible for the model to learn robust features, highlighting a challenge of extreme class imbalance.

  1. It is recommended that a discussion be held about the possible role of LLM and XAI in the future of this field.
  • Thank you for your feedback. The “Methods” sections, precisely under “Model Architecture” has been equipped with explainable AI (XAI). Under “Results”, explainable AI (XAI) subheading has been added, and image has been added depicting it for greater clarity.
  • A small section has also been added in Future Works.

“Furthermore, the integration of Large Language Models (LLMs) could revolutionize AI-assisted radiology. A multimodal LLM could synthesize CNN’s visual findings with patient history from electronic health records to generate preliminary, narrative-style reports, further reducing radiologist burden. ”

  1. In the method section, it is recommended that the formulas used be mentioned.
  • We have added key formulas for evaluation metrics (AUC, Sensitivity, Specificity, mAP) in the Methods
  • Area Under the ROC Curve (AUC):Calculated from the receiver operating characteristic plot.
  • Sensitivity (Recall):
  • Specificity:
  • Mean Average Precision (mAP):, where  is the average precision for class  at an Intersection-over-Union (IoU) threshold of 0.5.

N=represents the total number of distinct object classes (pathologies)

TP =True Positive

TN=True Negative

FP=False Positive

FN= False Negative

  1. If generative artificial intelligence was used in the writing of this manuscript, please disclose it.

Thank you for the suggestion on disclosure. We have added a section called GENERATIVE ARTIFICIAL INTELLIGENCE USE DISCLOSURE and clarified the following things. “During the preparation of this work, the authors used Grammarly to improve the readability, grammar, and language of parts of the manuscript. After using the tool, the authors reviewed and edited the content as needed and take full responsibility for the final manuscript's content and integrity.”

Regards and sincere thanks,

Authors

Reviewer 3 Report

Comments and Suggestions for Authors

This manuscript evaluates a lightweight convolutional neural network (CNN) trained using a hybrid dataset (public+local) to detect 14 thoracic abnormalities on multi-view chest X-rays (PA, AP, lateral) from a tertiary hospital in Nepal. The model incorporates preprocessing, augmentation, and Weighted Boxes Fusion to improve bounding-box accuracy. It is tested on 522 held-out images and achieves AUC 0.86, mean sensitivity 0.68, specificity 0.99, and mAP 0.93. It is further evaluated for PACS integration and offline deployment feasibility. The model demonstrates strong performance for common findings (consolidation, atelectasis) and acceptable robustness across poor-quality images and multiple views. The manuscript is clearly structured and addresses a clinically relevant gap in LMIC workflows.

This is a well written and well structured study. However, there are some points need to be addressed before its consideration for publication:

  1. Please clarify the CNN architecture. Provide explicit details regarding number of layers, backbone, training epochs, batch size, optimizer parameters, and loss functions. This is essential for reproducibility.
  2. Please expand on inter-reader variability during annotation. Three radiologists annotated the data, but the manuscript does not describe agreement statistics (e.g., Cohen’s kappa). This is important given the subjective nature of CXR labeling.
  3. Please define confidence thresholds used for performance metrics. mAP and sensitivity/specificity values depend heavily on threshold settings; clarification is needed.
  4. Please address class imbalance more explicitly. Rare classes (e.g., ILD, calcification) show low sensitivity. Discuss whether re-sampling, focal loss, or class-weighted training were explored.
  5. Please correct all figure citations. Figure 2, Figure 3, Figure 4 cited out of order. First appearances should follow numerical sequence. Ensure captions do not appear before first textual reference.
  6. Please provide more details on deployment evaluation. Describe number of PACS cases tested, radiologist feedback, and system latency.
  7. Please ensure that all abbreviations are defined at first use (e.g., WBF, OPD).
  8. Please specify whether lateral view performance metrics are pooled with frontal or evaluated separately.
  9. Please clarify whether pediatric images were completely removed before the train/val/test split.
  10. Please revise repeated phrasing in Results (e.g., “robustness,” “clinical utility”).
  11. Please add an error-analysis figure or short subsection summarizing typical false positives/false negatives.

Author Response

To,

Reviewer 3

Round 1

Thank you for providing your feedback. We agree with all points raised and have attempted to include them with amendments. The point-by-point response to comments are as follows:

This manuscript evaluates a lightweight convolutional neural network (CNN) trained using a hybrid dataset (public+local) to detect 14 thoracic abnormalities on multi-view chest X-rays (PA, AP, lateral) from a tertiary hospital in Nepal. The model incorporates preprocessing, augmentation, and Weighted Boxes Fusion to improve bounding-box accuracy. It is tested on 522 held-out images and achieves AUC 0.86, mean sensitivity 0.68, specificity 0.99, and mAP 0.93. It is further evaluated for PACS integration and offline deployment feasibility. The model demonstrates strong performance for common findings (consolidation, atelectasis) and acceptable robustness across poor-quality images and multiple views. The manuscript is clearly structured and addresses a clinically relevant gap in LMIC workflows.

This is a well written and well-structured study. However, there are some points that need to be addressed before its consideration for publication:

  1. Please clarify the CNN architecture. Provide explicit details regarding number of layers, backbone, training epochs, batch size, optimizer parameters, and loss functions. This is essential for reproducibility.
  • Thank you for the feedback. The above specifications have been mentioned in Methods section.

Preprocessing and Augmentation

All radiographs were resized to 640 × 640 pixels and normalized using a two-step strategy: (i) Min–max normalization to scale pixel intensities to [0,1]. (ii) Z-score normalization using dataset-level mean and standard deviation.

Extensive augmentation was applied using Albumentations, including random rotations (±15°), horizontal flips, contrast and brightness adjustment, gamma correction, Gaussian noise, Gaussian blur, CLAHE, and sharpening. Medical-specific artifact simulations included overlaying synthetic text labels, ECG wires, and ICU bed-like structures. Bounding box integrity was preserved with minimum visibility ≥0.3 and minimum area ≥1 pixel.

Model Architecture

A computationally efficient, lightweight convolutional neural network (CNN) architecture was developed to enable deployment in resource-constrained clinical environments. The backbone network was a custom Cascade Multi-Resolution Feature Network (CMRF-Net) consisting of ten sequential stages: input normalization using Group Normalization; a Shallow Gradient Extractor with dual 3 × 3 convolutions and LeakyReLU activations; a Dual-Path Expansion Unit comprising parallel convolutional branches with channel fusion; a Hierarchical Aggregation Stack implemented with RepConv blocks; a Progressive Depth Encoder using depthwise separable convolutions with spatial downsampling; a Multi-Scale Bottleneck Cluster incorporating spatial pyramid pooling; a Feature Lift and Redistribution module enabling top–down multi-scale feature fusion; a Secondary Upscale and Multi-Branch Mixing module; a Downscale Reintegration Block; and a Deep Recombination Cluster based on dilated convolutions using atrous spatial pyramid pooling (ASPP). Final predictions were produced through YOLO-style resolution-aligned output heads together with a global image-level classification head, enabling simultaneous multi-scale localization and whole-image abnormality assessment.

  1. Please expand on inter-reader variability during annotation. Three radiologists annotated the data, but the manuscript does not describe agreement statistics (e.g., Cohen’s kappa). This is important given the subjective nature of CXR labeling.
  • Thank you for the feedback. The following statement has been added in Methods section in Evaluation Metrics.

“Image quality was evaluated through expert review and objective metrics. During annotation, radiologists identified images with severe motion blur, exposure issues, or positioning errors. We also used Signal-to-Noise Ratio (SNR) and Contrast-to-Noise Ratio (CNR) to flag potentially inadequate images. Those falling below our quality thresholds were reviewed by a senior radiologist, and diagnostically unacceptable images were excluded. Inter-rater agreement was quantified using Cohen's kappa (κ). Model performance was then rigorously assessed using comprehensive metrics. We conducted a reader study to evaluate clinical utility, measuring the AI system's integration ease with radiologists' workflow.”

  1. Please define confidence thresholds used for performance metrics. mAP and sensitivity/specificity values depend heavily on threshold settings; clarification is needed.
  • Thank you for the feedback. This has been added in Methods Section in Evaluation Metrics.

“For the calculation of sensitivity, specificity, and precision, a confidence threshold of 0.5 was applied to the model's predictions. The mean Average Precision (mAP) was calculated at an Intersection over Union (IoU) threshold of 0.5 (mAP@0.5), which is standard for object detection tasks.”

  1. Please address class imbalance more explicitly. Rare classes (e.g., ILD, calcification) show low sensitivity. Discuss whether re-sampling, focal loss, or class-weighted training were explored.
  • Thank you for the feedback. This has been added to the Results

“The pronounced class imbalance in our dataset (e.g., Calcification: 0.38%, Other Lesion: 0.95%) presented a significant challenge. We employed a class-weighted loss function during training to penalize misclassifications of rare classes more heavily and used targeted augmentation (e.g., copy-paste augmentation for rare findings) on the training set. However, for extremely rare classes like "Other Lesion", these measures were insufficient to overcome the lack of representative examples, resulting in poor recall.”

  1. Please correct all figure citations. Figure 2, Figure 3, Figure 4 cited out of order. First appearances should follow numerical sequence. Ensure captions do not appear before first textual reference.
  • Thank you for your feedback. This has been addressed throughout manuscript.

  1. Please provide more details on deployment evaluation. Describe number of PACS cases tested, radiologist feedback, and system latency.
  • Thank you for your feedback. This has been addressed in Deployment Testing in Methods section.

“The system was integrated with the hospital PACS and tested in a silent clinical trial on 100 consecutive anonymized studies. Three radiologists used the system for one week and provided structured feedback via 5-point Likert scales.

The model was deployed through an offline-capable React/NodeJS web interface running locally with edge-GPU acceleration. Average end-to-end latency (DICOM retrieval → AI overlay display) was recorded. All AI-generated bounding boxes and confidence scores were exportable as DICOM-compatible overlays, ensuring full interoperability with existing radiology workflows.”

  1. Please ensure that all abbreviations are defined at first use (e.g., WBF, OPD).
  • Thank you for your feedback. This has been addressed throughout manuscript.
  1. Please specify whether lateral view performance metrics are pooled with frontal or evaluated separately.
  • Thank you for your suggestion. Lateral performance metrics are pooled in with frontal view performance metrics. This has been now mentioned in Results section.
  1. Please clarify whether pediatric images were completely removed before the train/val/test split.
  • Thank you for your suggestion. We agree and have mentioned that pediatric images were completely removed from datasets before split and have added clear methodology with defined inclusion/exclusion criteria in Methods section.

Study Design and Datasets
This retrospective study developed and evaluated a deep learning system for automated detection and localization of thoracic abnormalities on chest radiographs. The model was first pretrained on two public datasets: the Vin Big Chest X-ray dataset (5,500 images) and the NIH ChestX-ray dataset (1,000 images). Domain-specific fine-tuning was conducted using adult chest radiographs acquired from the Emergency Department, Intensive Care Unit, and Outpatient Clinics of Tribhuvan University Teaching Hospital (TUTH), Nepal, between January 1, 2024, and January 1, 2025.

Inclusion criteria were: (i) age ≥18 years, (ii) PA, AP, or lateral digital chest radiographs, and (iii) studies performed in ER, ICU, or OPD. Exclusion criteria were: (i) pediatric patients, (ii) post-operative or post-interventional radiographs with extensive hardware artifacts, and (iii) severely corrupted or non-diagnostic images. Pediatric cases were excluded to avoid anatomical confounding.

  1. Please revise repeated phrasing in Results (e.g., “robustness,” “clinical utility”).
  • Thank you for your feedback. This has been amended.
  1. Please add an error-analysis figure or short subsection summarizing typical false positives/false negatives.
  • Thank you for your feedback. We have added a new subsection 5 Error Analysisin the Results, summarizing common false positives/negatives.

“We conducted an analysis of the model's errors on the test set to identify common failure modes. The most frequent false positives occurred for 'Lung Opacity' and 'Infiltration,' often in cases with prominent vascular markings or suboptimal exposure that the model misinterpreted as pathology. The most frequent false negatives were for 'Pneumothorax', predominantly involving small, apical, or lateral pneumothoraxes on portable AP films, and for the 'Other Lesion' class, where the limited number of examples hindered learning.”

Regards and sincere thanks,

Authors

Round 2

Reviewer 1 Report

Comments and Suggestions for Authors

The authors have addressed all the comments. The manuscript can be accepted in its current version

Reviewer 2 Report

Comments and Suggestions for Authors

I thank the authors of this manuscript for their efforts in addressing the peer reviewers' comments.